# Explainable Artificial Intelligence for Human Decision Support System in the Medical Domain

**Samanta Knapič** [1,2,*] **, Avleen Malhi** [1,3] **, Rohit Saluja** [1,4] **and Kary Främling** [1,2]

1 Department of Computer Science, Aalto University, Konemiehentie 2, 02150 Espoo, Finland;
  avleen.malhi@aalto.fi (A.M.); rohit.saluja@aalto.fi (R.S.); kary.framling@cs.umu.se (K.F.)
2 Department of Computing Science, Umeå University, 90187 Umeå, Sweden
3 Department of Computing and Informatics, Bournemouth University, Poole BH12 5BB, UK
4 Department of Electrical Engineering and Computer Science, KTH Royal Institute of Technology,
  11428 Stockholm, Sweden
* Correspondence: samanta.knapic@aalto.fi

**Abstract:** In this paper, we present the potential of Explainable Artificial Intelligence methods for decision support in medical image analysis scenarios. Using three types of explainable methods applied to the same medical image data set, we aimed to improve the comprehensibility of the decisions provided by the Convolutional Neural Network (CNN). In vivo gastral images obtained by a video capsule endoscopy (VCE) were the subject of visual explanations, with the goal of increasing health professionals' trust in black-box predictions. We implemented two post hoc interpretable machine learning methods, called Local Interpretable Model-Agnostic Explanations (LIME) and SHapley Additive exPlanations (SHAP), and an alternative explanation approach, the Contextual Importance and Utility (CIU) method. The produced explanations were assessed by human evaluation. We conducted three user studies based on explanations provided by LIME, SHAP and CIU. Users from different non-medical backgrounds carried out a series of tests in a web-based survey setting and stated their experience and understanding of the given explanations. Three user groups ($n = 20, 20, 20$) with three distinct forms of explanations were quantitatively analyzed. We found that, as hypothesized, the CIU-explainable method performed better than both LIME and SHAP methods in terms of improving support for human decision-making and being more transparent and thus understandable to users. Additionally, CIU outperformed LIME and SHAP by generating explanations more rapidly. Our findings suggest that there are notable differences in human decision-making between various explanation support settings. In line with that, we present three potential explainable methods that, with future improvements in implementation, can be generalized to different medical data sets and can provide effective decision support to medical experts.

**Keywords:** explainable artificial intelligence; human decision support; image recognition; medical image analyses

## 1. Introduction

In conventional diagnostics, possible lesions in captured images are checked manually by a doctor in a medical setting. This manual approach is time-consuming and relies on the prolonged attention of the doctor, who has to examine thousands of images from a single medical procedure. On the other hand, in recent years, deep learning and AI-based extraction of information from images have received growing interest in fields such as medical diagnostics, finance, forensics, scientific research and education. In these domains, it is often necessary to understand the reason for the model's decisions so that the human can validate the decision's outcome [1].

Recently, a number of computer-aided diagnostic (CAD) tools have been undergoing development to allow for the automated or semi-automated identification of lesions.

By automatically extracting features, the newly introduced Convolutional Neural Network (CNN), also known as a deep neural network, has been able to produce results with significantly higher accuracy compared to standard approaches [2]. Reinforcement learning techniques and deep learning methods trained on massive data sets have surpassed the efficiency of humans, producing impressive results even in the medical field. Through the use of machine learning techniques, the lesion detection method can be automated with promising accuracy, saving both time and manual effort [3]. Well-trained machine learning systems have the ability to generate accurate predictions regarding various anomalies and can therefore be used as effective clinical practice tools. However, although their core mathematical concepts can be understood, they lack an explicit declarative information representation and have difficulty producing the underlying explanatory structures [3].

As AI becomes more effective and is being used in more sensitive circumstances with significant human implications, trust in such systems is becoming increasingly essential [1]. Humans must be able to understand, reproduce and manipulate machine decision-making processes in real-time. As a result, there is an increasing need to improve the comprehensibility of decisions made by machine learning algorithms so that they can be replicated in real applications, especially in medicine. This requires systems that produce straightforward, understandable and explainable decisions, along with re-traceable results on demand. As medical professionals work with dispersed, heterogeneous and complex data sources, Explainable Artificial Intelligence (XAI) can help to promote the application of AI and machine learning in the medical sector, and in particular, it can help to foster transparency and trust. Therefore, before they can be trusted, machine learning models should be able to justify their decisions. Explanation support will help to clarify the decisions made by a black-box model, making it more intuitive for humans. Additional explanation of the decisions made can increase the reliability of the method and thus assist medical professionals in making the correct diagnosis [4] (as shown in Figure 1).

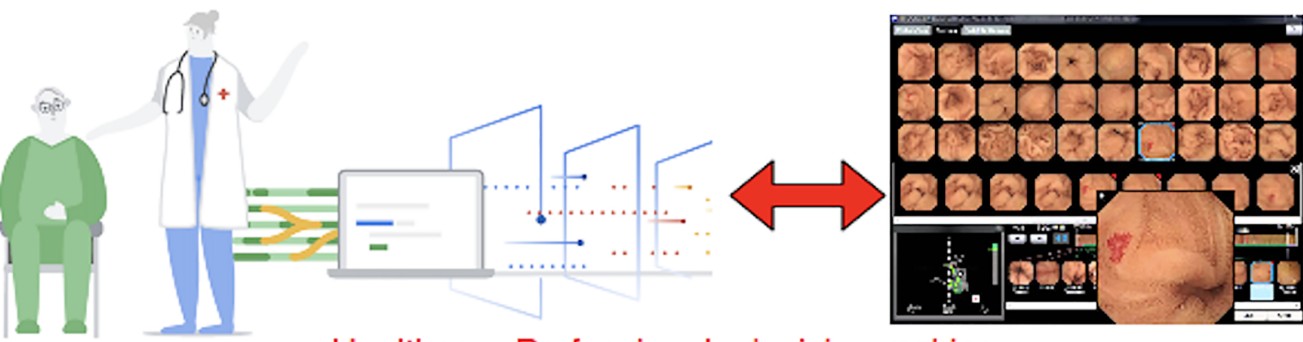

**Figure 1.** XAI helping medical professionals in decision-making.

In the present study, we introduced a neural network, particularly a Convolutional Neural Network (CNN), with a specific application in the field of medicine. We aimed to improve the comprehensibility of the decisions provided by the CNN by implementing two post hoc interpretable machine learning methods, called Local Interpretable Model-Agnostic Explanations (LIME) and SHapley Additive exPlanations (SHAP), and an alternative explanation approach, the Contextual Importance and Utility (CIU) method, to explain machine learning predictions [5]. With the aim of helping health professionals to trust black-box predictions, we applied the explanations to in vivo gastral images that were obtained by video capsule endoscopy (VCE). The three explanation types were applied to the same medical image data set and then assessed by human evaluation. We conducted preliminary human decision support user studies to determine how well humans can understand the provided explanations and to examine the effect of the explanations on human decision-making. The three user groups were presented with the decision support of three distinct explainable methods: LIME, SHAP and CIU, which automatically generate

different visually based explanations. The following were our research questions: RQ1: Will Explainable Artificial Intelligence improve the trustworthiness of AI-based computer vision systems in the medical domain? RQ2: Can various XAI approaches be used as a human decision support system, and can their explanation strategies be compared? RQ3: Can users recognize the effectiveness of the generated explanations?

## 2. Literature Review

Although machine learning models can be considered reliable, the effectiveness of these systems is limited by the current inability of machines to explain their decisions and actions to human users. While there is an increasing body of work on interpretable and transparent machine learning algorithms, the majority of studies are primarily centered on users with technical knowledge. A recently published paper provides a comprehensive survey of Explainable Artificial Intelligence studies [6]. Guidotti et al. [7] conducted a detailed analysis of XAI approaches for describing black-box models, and Anjomshoae et al. [8] provided a systematic review of the literature on explainable agents. By critically discussing the notions related to the concept of explainability as well as the evaluation approaches for XAI methods Vilone et al. [9] identify future research directions with explainability as the starting component of any artificial intelligence system. Contextual utility and the importance of features [10,11] have been the basis of an early approach to understanding the decisions of ML models, but with the emergence of deep learning as a popular data analysis tool, new approaches such as SHAP (SHapley Additive exPlanations) [12], LIME (Local Interpretable Model-Agnostic Explanations) [3], CIU (Contextual Importance and Utility) [11], ELI5 [13], VIBI and L2XSkater [14] have been developed to explain the decisions of machine learning models.

### 2.1. Explainable Artificial Intelligence in Machine Learning

Xie et al. [15] provided a guide to explainability within the realm of deep learning by discussing the characteristics of its framework and introducing fundamental approaches that lead to explainable deep learning. Samek et al. [16] researched a deep learning method for image recognition with an explainability approach, and they proposed two methods for explaining sensitivity to input changes. Choo et al. [17] presented another insightful perspective on potential directions and emerging problems in explainable deep learning. They discussed implementation possibilities concerning human interpretation and control of deep learning systems, including user-driven generative models, progression of visual analytics, decreased use of training sets, improved AI robustness, inclusion of external human intelligence and deep learning visual analytics with sophisticated architectures. In a project launched by DARPA [18], the researchers provided simple conceptual and example applications on the current state of work on Explainable Artificial Intelligence in the domains of defense, medicine, finance, transportation, military and legal advice. In [19], a machine vision-based deep learning explainable framework was used to investigate plant stress phenotyping; feature maps and unsupervised learning were used with approximately 25,000 photos to calculate stress intensity. Hase and Bansal [20] presented an example of human subject tests and studied the impact of algorithmic explanations on human decision-making. Their studies were the first to provide precise estimates of how explanations impact simulatability across a broad spectrum of data domains and explanation techniques. They demonstrated that criteria for measuring interpretation methods must be carefully chosen and that existing methods have considerable room for development.

### 2.2. Explainable Artificial Intelligence in the Medical Field

Over the past few years, AI-based image information retrieval has generated considerable interest in medical diagnostics. Holzinger et al. [21] emphasized the importance of using Explainable AI in the medical field to assist medical practitioners in making decisions that are explainable, transparent and understandable. They predicted that the ability to explain the machine learning decision would support the adoption of machine learning

in the medical field. In another article [22], in the context of an application task in digital pathology, Holzinger et al. address the importance of making decisions straightforward and understandable with the use of Explainable Artificial Intelligence. Sahiner et al. [23] outline the past and present state of deep learning research in medical imaging and radiation therapy, address challenges and their solutions, and conclude with future directions. Amann et al. [24] investigated the function of XAI in clinical settings and drew the conclusion that in order to eliminate challenges to ethical principles, the inclusion of explainability is an important requirement. This can help to ensure that patients remain at the center of treatment and can make knowledgeable and independent decisions about their well-being with the help of medical professionals. Ribeiro et al. [25] developed an explanation technique called Local Interpretable Model-agnostic Explanations (LIME) as a means to explain the predictions of the classifier in a reliable and interpretable way. The model's versatility was shown by including text and image explanations for a variety of models. In making decisions between models, it supported both expert and non-expert users while evaluating their confidence and improving untrustworthy models by providing insight into their predictions. Further, in [1], the authors address the effect of explainability on trust in AI and computer vision systems through the improved understandability and predictability of deep learning-based computer vision decisions on medical diagnostic data. They also explore how XAI can be used to compare the recognition techniques of two deep learning models: Multi-Layer Perceptron and Convolutional Neural Network (CNN).

A neural network called PatchShuffle Stochastic Pooling Neural Network (PSSPNN) provides accurate classification results in the diagnosis of subjects suffering from secondary pulmonary tuberculosis or pneumonia versus healthy subjects. Wang et al. [26] proposed a novel stochastic pooling neural network (SPNN) inspired by the architecture of VGG-16 along with a PatchShuffle regularization term. Here, Grad-CAM was also used to explain and examine the quality of the results. This was coupled with an improved multi-way data augmentation technique (to avoid overfitting) to outperform nine state-of-the-art neural network architectures in terms of F1 score over 10 runs.

Another novel neural network architecture is Attention Network (ANC), which was recently proposed for the identification of COVID-19 based on the classification of lung images. ANC was used by Zhang et al. [27] with a proposed 18-way data augmentation method to avoid overfitting, and the results were explained using Grad-CAM. The authors claimed that ANC 9 performed better than nine other state-of-the-art neural networks on the particular task according to seven metrics, including accuracy and F1 score. However, the results were conducted with a very small image data set.

## 3. Background

Recently, laws and regulations have been moving towards transparency requirements for information systems to prevent unintended adverse effects in decision-making. In particular, the General Data Protection Regulations (GDPR) of the European Union grant users the right to be informed regarding machine-generated decisions [28]. Consequently, individuals who are affected by decisions made by a machine learning-based system may seek to understand the reasons for the system's decision outcome.

### 3.1. Black-Box Predictions

Like we cannot yet look into someone's brain to examine its thoughts and mindset, we also do not yet have access to the internal processing of the deep neural network [29]. Therefore, the main concerns about machine learning models' decisions are whether we should trust these decisions and how machine learning or deep learning models make their decisions. Relevant principles in relation to these questions, which refer to the ability to observe the processes that lead to decision making within the model, are listed below.

1.  Transparency: A model is considered transparent if it is understandable on its own, which usually applies to easily interpretable models [30]. Simpler machine learning

models tend to be more transparent and are thus inherently more interpretable due to their simple structure, such as models built with linear regression.

2.  Interpretability: The ability to describe or provide meaning that is clear to humans is known as interpretability. Models are considered interpretable if they are described in a way that can be further explained, such as through domain awareness [31]. The concept behind interpretability is that the more interpretable a machine learning system is, the easier it is to define cause–effect relationships between the inputs and outputs of the system [32].

3.  Explainability: Explainability is more closely linked to the machine learning system's dynamics and internal logic. While a model is training or making decisions, the more explainable it is, the greater the human understanding of the internal procedures [32].

An interpretable model does not imply that humans can comprehend its internal logic or underlying processes; thus, interpretability does not necessarily imply explainability, and vice versa. Interpretability alone is not sufficient, as the presence of explainability is also important. To meet these objectives, a new area known as XAI (Explainable Artificial Intelligence) has arisen, with the goal of developing algorithms that are both efficient and explainable.

### 3.2. Explainable Artificial Intelligence (XAI)

As depicted in Figure 2, Explainable Artificial Intelligence (XAI) methods have been developed in order to achieve greater transparency and produce explanations for AI systems. The Explainable Artificial Intelligence (XAI) research area, as a developing branch of artificial intelligence (AI), is investigating various approaches that will allow the behavior of intelligent autonomous systems to be interpretable and understandable to humans. Human–machine interaction, on the bridge between Data Science and Social Sciences, is leading to more advanced AI and, at the same time, contributing to more transparent, reasonable and thus responsible AI. In general, explanations help evaluate the strengths and the limitations of a machine learning model, thereby facilitating trustworthiness and understandability [16,18,33]. Post hoc explanations are one approach to extracting the information on a black-box model's process of reaching a certain decision. They can provide useful information, particularly for practitioners and end users who are interested in instance-specific explanations rather than the internal workings of the model. The goal of XAI models is to create explainable models while maintaining high learning efficiency (prediction accuracy) [1].

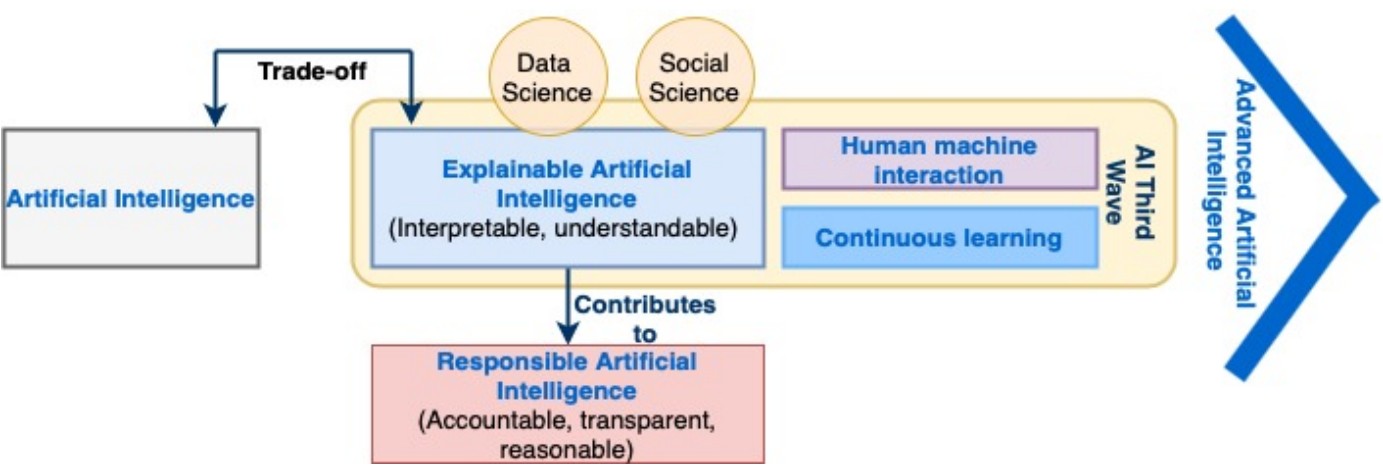

**Figure 2.** Basic concepts of XAI.

### 3.3. LIME, SHAP and CIU

Several post hoc explanation tools for explaining a specific model prediction, including LIME, SHAP, L2X and VIBI, have been proposed. One example of a post hoc tool, shown

in Figure 3, is Local Interpretable Model-agnostic Explanations (LIME), which explains a model's prediction by using the most important contributors. LIME approximates the prediction locally by perturbing the input around the class of interest until it arrives at a linear approximation [25] and helps the decision-maker in justifying the model's behavior. SHapley Additive exPlanations (SHAP) is another example that describes the outcome by "fairly" distributing the prediction value among the features, depending on how each function contributes [34]. The attributes are as follows: (i) global interpretability—the importance of each indicator that has a positive or negative impact on the target variable; (ii) local interpretability—SHAP values are determined for each instance, significantly increasing transparency and aiding in explaining case prediction and major decision contributors; and (iii) SHAP values can be calculated for any tree-based model [4]. Both of the above approaches approximate the local behavior of a black-box system with a linear model. Therefore, they only provide local fidelity, and faithfulness to the original model is lost. As an alternative, the Contextual Importance and Utility (CIU) method for explaining machine learning predictions is based on the notion that the importance of a feature depends on the other feature values; thus, a feature that is important in one context might be irrelevant in another. The feature interaction allows for the provision of high-level explanations, where feature combinations are appropriate or features have interdependent effects on the prediction. The CIU method consists of two important evaluation methods: (1) Contextual Importance (CI), which approximates the overall importance of a feature in the current context, and (2) Contextual Utility (CU), which provides an estimation of how favorable or unfavorable the current feature value is for a given output class [8]. More specifically, the CU provides insight into how much a feature contributes to the current prediction relative to its importance, and alongside the feature importance value, the utility value adds depth to the explanation.

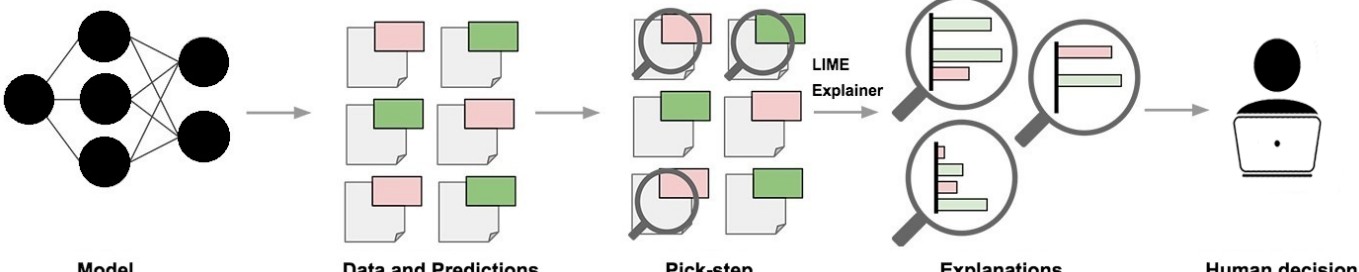

**Figure 3.** Pipeline of post hoc explainable tool LIME.

## 4. XAI Methods

### 4.1. LIME

Our first post hoc explainability algorithm is Local Interpretable Model-agnostic Explanations (LIME). In [25], Ribeiro et al. proposed a local surrogate model, the LIME method, developed to help users by generating explanations for a black-box model's decisions in all instances. LIME's explanation is based on evaluating the classifier model's behavior in the vicinity of the instance to be explained on the basis of local surrogate models, which can be linear regressions or decision trees, as observed in Equation (1). Here, $x$ is an instance being explained. The explanation of $x$ is the result of the maximization of the fidelity term $\rfloor(f, g, \pi_x)$ with complexity of $\Omega(g)$. $f$ represents a black-box model, which is explained by an explainer, represented by $g$. The local surrogate model tries to match the data in the vicinity of the prediction that needs to be explained. Fitting the local model requires sufficient data in the vicinity of the instance being explained, which is carried out by sampling data from its neighborhood.

$$explanation(x) = argmin_{g \in G} \rfloor(f, g, \pi_x) + \Omega(g) \tag{1}$$

Initially, LIME used a perturbation technique [25] to generate samples from the original data set. However, in R [35] and Python [36] implementations, it adopted a different approach. In their implementation, the univariate distribution of each feature is considered to estimate distributions for each feature, and categorical and numerical features are treated differently. For categorical features, sampling is based on probabilities of the frequency of each category. However, for numerical features, there are three alternatives: First, the original data set is grouped into bins based on its quantiles, and one bin is randomly picked and sampled uniformly between the minimum and maximum of the selected bin. Second, LIME approximates the original distribution of numerical features through a normal distribution, and the approximated distribution is used to sample the data for that feature. Third, the actual distribution of numerical features is approximated using a kernel density function, from which data are sampled. LIME utilizes an exponential kernel by design, with the kernel width equal to the square root of the number of features.

*4.2. SHAP*

For the second human evaluation user study, we used Shapley Additive exPlanations (SHAP) as our second post hoc explainability algorithm to generate explanations. We examined Deep SHAP Explainer and SHAP Gradient Explainer, which combine ideas from Integrated Gradients (which require a single reference value to integrate from), SHapley Additive exPlanations (SHAP) and SmoothGrad (which averages gradient sensitivity maps for an input image to identify pixels of interest) into a unified expected value equation. For the current study, we chose the Kernel SHAP algorithm, which is a model-agnostic method for estimating SHAP values for just about any model. The SHAP KernelExplainer works for all models but is slower than the other model type-specific algorithms, as it makes no assumptions about the model type. It provided the best results for us, despite being slower than other explainers and providing an approximation rather than exact SHAP values. The Kernel SHAP algorithm is based on Lundberg et al.'s paper [34] and builds on the open-source Shap library from their first paper [34].

SHAP [34] aims to explain individual predictions by employing the game-theoretic Shapley value [37], as shown in Figure 4. This approach uses the concept of coalitions in order to compute (as shown in Equation (2)) the Shapley value of features for the prediction of instance $(x)$ by the black-box model $(f)$. The average marginal contribution $(\phi_j^m)$ of feature $(j)$ in all possible coalitions is the Shapley value. The marginal contribution is calculated as in Equation (3), where $\hat{f}(x_{+j}^m)$ and $\hat{f}(x_{-j}^m)$ are predictions of black-box $f$ without and with replacement of the $j^{th}$ feature of instance $x$ from the sample.

$$\phi_j(x) = \frac{1}{M} \sum_{m=1}^{M} \phi_j^m \qquad (2)$$

$$\phi_j^m = \hat{f}(x_{+j}^m) - \hat{f}(x_{-j}^m) \qquad (3)$$

**Figure 4.** Pipeline of post hoc explainable tool SHAP.

### 4.3. CIU

For the third human evaluation user study, we implemented the CIU method. The Contextual Importance and Utility (CIU) method explains the model's outcome not only based on the degree of feature importance, but also based on the utility of features (usefulness for the prediction) [11]. It consists of two important evaluation methods: (1) Contextual Importance (CI), which approximates the overall importance of a feature in the current context, and (2) Contextual Utility (CU), which provides an estimation of how favorable or unfavorable the current feature value is for a given output class.

CIU differs radically from LIME and SHAP because it does not create or use an intermediate surrogate model or make linearity assumptions [11]. Here, Contextual Importance (CI) and Contextual Utility (CU) are used for generating the explanation and interpretation based on the contributing features of the data set. It also helps to justify why one class is preferred over another. These explanations have *contextual* capabilities, which means that one feature may be critical for making a decision in one situation while being irrelevant in another. The mathematical definitions (detailed in [38]) of CI and CU are given in Equations (4) and (5), respectively.

$$CI_j(\vec{C}, \{i\}) = \frac{cmax_j(\vec{C}, \{i\}) - cmin_j(\vec{C}, \{i\})}{absmax_j - absmin_j} \tag{4}$$

$$CU_j(\vec{C}, \{i\}) = \frac{out_j(\vec{C}) - cmin_j(\vec{C}, \{i\})}{cmax_j(\vec{C}, \{i\}) - cmin_j(\vec{C}, \{i\})} \tag{5}$$

Here, $CI_j(\vec{C}, \{i\})$ is the contextual importance of a given set of inputs $\{i\}$ for a particular output $j$ in the context $\vec{C}$. $absmax_j$ is the maximum possible value for output $j$, and $absmin_j$ is the minimum possible value for output $j$. $cmax_j(\vec{C}, \{i\})$ is the maximum value of output $j$ observed when modifying the values of input $\{i\}$ and retaining the values of the other inputs at those specified by $\vec{C}$. Correspondingly, $cmin_j(\vec{C}, \{i\})$ is the minimum value of output $j$. Similarly, for contextual utility $CU_j(\vec{C}, \{i\})$, $out_j(\vec{C})$ is the value of output $j$ for context $\vec{C}$. Figure 5 shows an example of obtaining CI and CU values for one input–output pair.

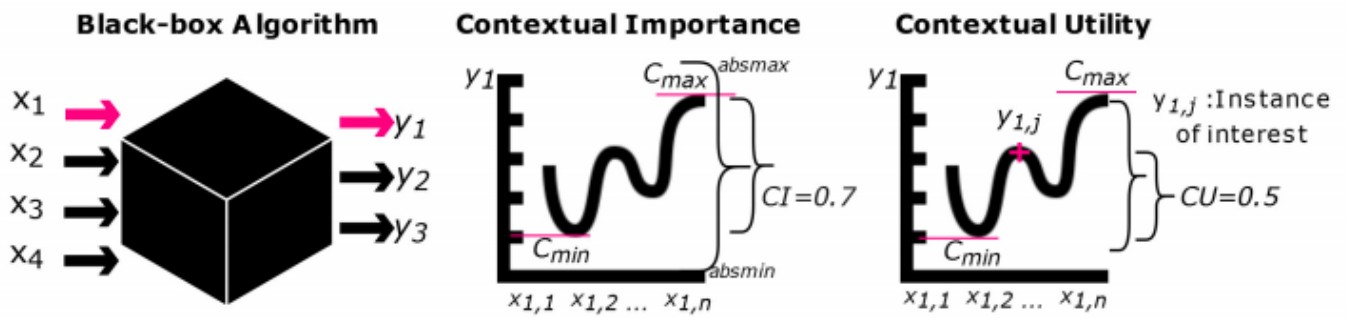

**Figure 5.** CIU explainable method [5].

### 4.4. Grad-CAM

Gradient-weighted Class Activation Mapping (Grad-CAM) is another very promising post hoc explainability method that is useful for producing visual explanations in classification tasks performed with any Convolutional Neural Network (CNN). Grad-CAM is reported to be highly effective with respect to both interpretability and faithfulness to the underlying CNN. A limitation of Grad-CAM is that it is an explainability method focused only on computer vision tasks, whereas SHAP, LIME and CIU can be used for textual and tabular data as well [39]. Grad-CAM shows immense promise but was not considered in this study due to a paucity of time and resources .

The three XAI techniques implemented and compared in this research are LIME, SHAP and CIU. A qualitative comparison of the three techniques is shown in Table 1.

**Table 1.** Comparison of LIME, SHAP and CIU [40,41].

| Method | Advantages | Disadvantages |
| --- | --- | --- |
| SHAP | The greatest advantages of an explainability technique such as SHAP are its solid fundamental roots in game theory. This ensures that the explanation of a prediction instance is fairly distributed among the features. | SHAP is a slow and computationally expensive explainability technique as it requires Shapley values to be calculated for various features in a prediction instance. This also makes SHAP impractical for calculating global explanations if there are a lot of prediction instances. This is particularly true for Kernel SHAP. |
| LIME | As LIME builds a local surrogate model, it offers the flexibility to replace the underlying machine learning model while using the same surrogate model. For example, if the audience of the generation understands decision trees the best, the underlying ML can be changed, but the explanations can still serve as decision trees. | One of the greatest disadvantages of LIME is that the explanations provided can be really unstable. LIME samples data points from a Gaussian distribution, and this introduces some randomness to the process of producing explanations. If the sampling process is repeated a sufficient number of times, it leads to different explanations for a single prediction instance. This reduces trust in the explanation. |
| CIU | The greatest advantage of CIU is that it does not rely on a surrogate model, which allows it to provide more detailed, transparent and stable explanations as compared to additive feature attribution-based methods while remaining more lightweight. This makes the model much faster to run as compared to LIME and SHAP. | CIU is still in the early stage of development as compared to additive feature attribution methods such as LIME and SHAP. |

## 5. Methodology

This section summarizes the methods used to assess the effect of explanations on human decision-making. The entire process is divided into four parts, as shown in Figure 6: data preprocessing; CNN model application; LIME, SHAP, and CIU explanation generation; and assessment of human decision-making in the form of user studies. First, we generated predictions made by a machine learning model using the selected data set. In addition, we implemented explanations of three different explainable methods. The methods responsible for assisting human decision making are Local Interpretable Model-agnostic Explanations (LIME), SHapley Additive exPlanations (SHAP) and Contextual Importance and Utility (CIU). We then evaluated how adding explanations with LIME, SHAP or CIU affect human decision making by conducting a user study.

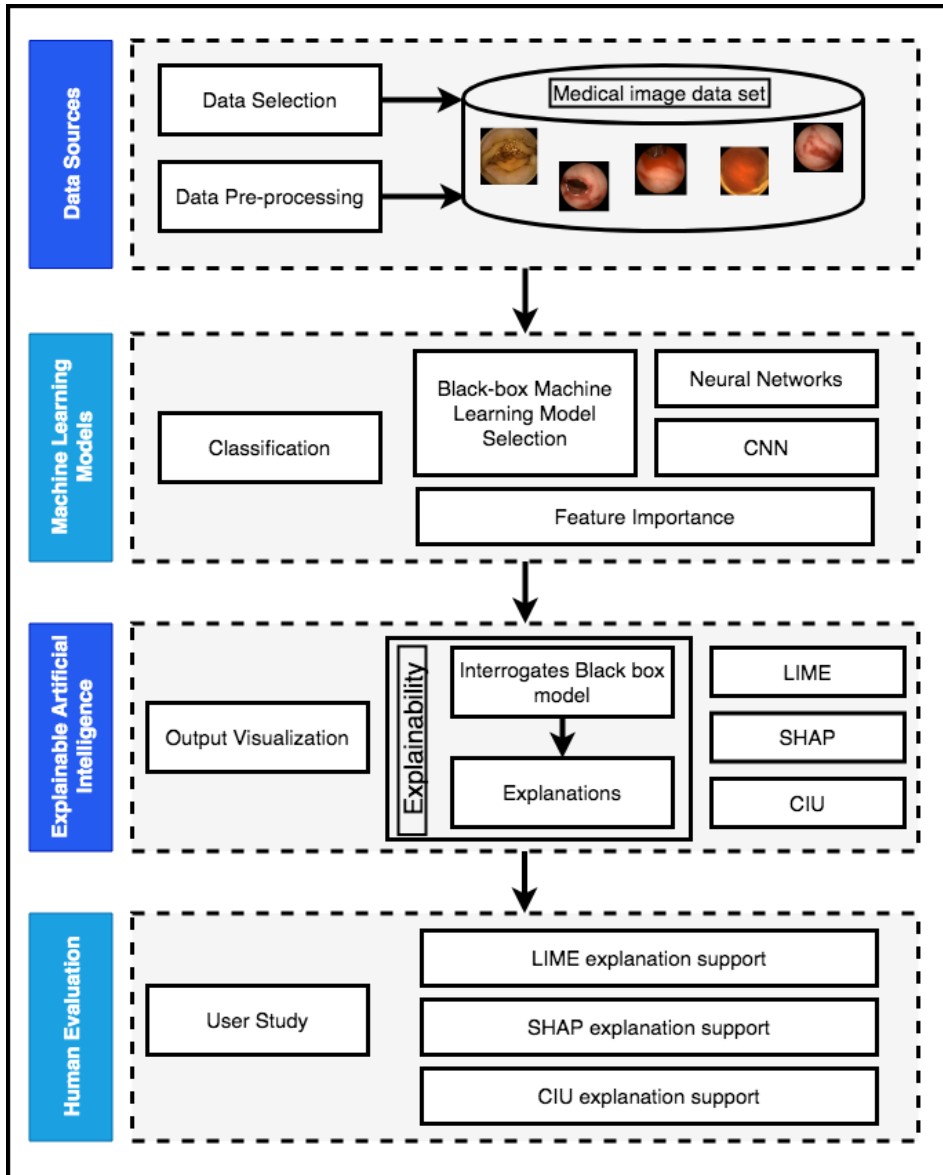

**Figure 6.** Workflow of the proposed method.

*5.1. Image Data Set*

The medical data set considered in the present study was generated by a Video Capsule Endoscopy (VCE), a noninvasive procedure to visualize a patient's entire gastroenterological tract. The aim of the VCE procedure is to detect segments of red lesions in the small bowel, one of the major organs where bleeding from unknown causes occurs, in order to detect signs of bleeding or polyps. There has been a major breakthrough in diagnosing small bowel diseases with VCE; however, the problem is that a single examination produces 10 h of video material, which requires a long time to read. As a result, analytical methods are needed to improve the efficiency and accuracy of the diagnosis. The 3295 images in the Red Lesion Endoscopy data set are publicly available and were retrieved from Coelho (https://rdm.inesctec.pt/dataset/nis-2018-003) [2] accessed on 10 September 2020. The data set includes two sets of images. The first set contains 1131 images with lesions and 2164 without lesions, for a total of 3295 images. The second set contains 439 images with lesions and 151 without lesions, for a total of 600 images. Both sets also contain manually annotated masks marking the bleeding area in each image. All lesions were annotated manually and approved by a trained physician. We focused on Set 1, with a total of 3295 images, of which 10% were used for testing and 90% were used for training.

### 5.2. Implementation of the Black-Box Model

To begin, we split our data and labels into training and validation sets (randomly assigned). The images are representative of the medical application situation, as shown in Table 2, and include both bleeding and non-bleeding (normal) examples. All 3295 images were resized to 150 × 150 pixels for faster and more accurate computation. A CNN (convolutional neural network) model with 50 epochs and a batch size of 16 was used to train the data set (shown in Figure 7), achieving a validation accuracy of 98.58%, as shown in Figure 8. We trained our CNN model based on labels assigned to each image to recognize the bleeding versus normal (non-bleeding) medical images. The labels were made using the repository's [2] annotated images as a reference point.

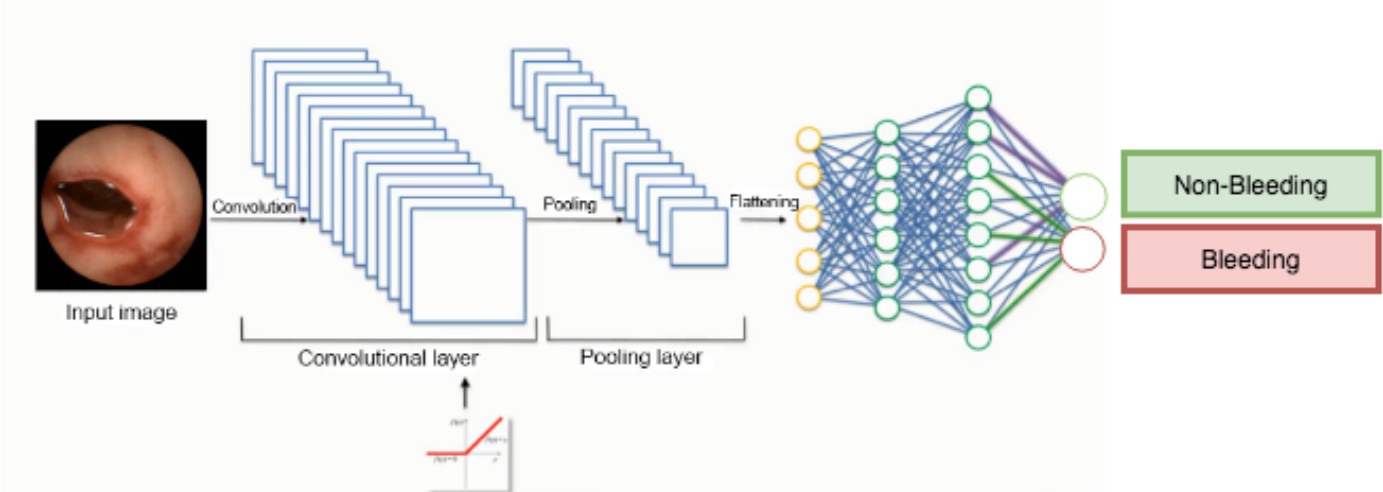

**Figure 7.** CNN model.

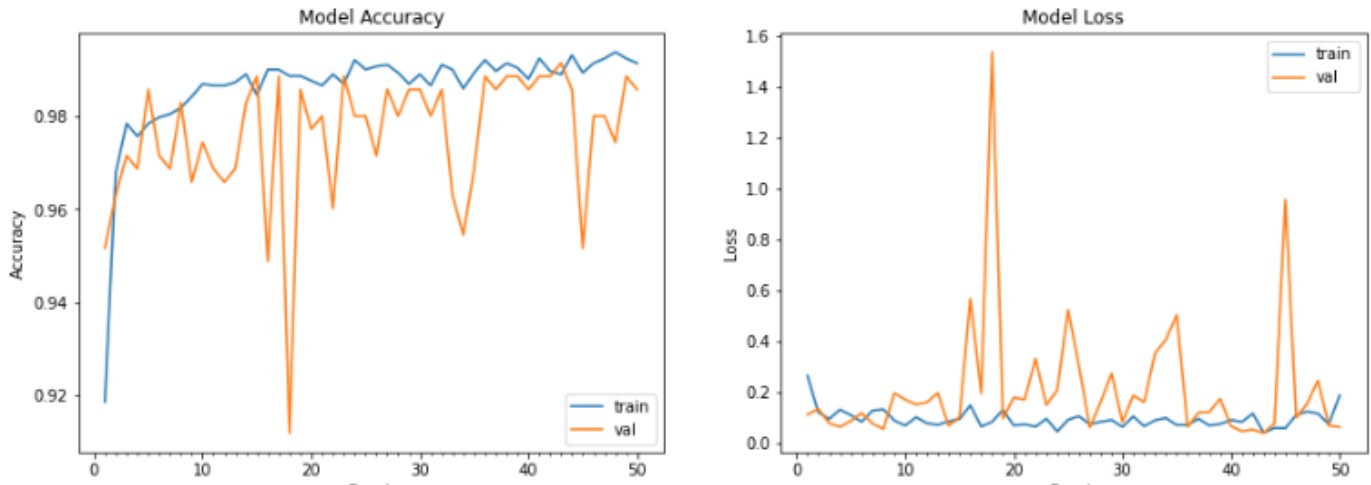

**Figure 8.** Model's accuracy and loss.

**Table 2.** Normal (non-bleeding) and bleeding images used for training and testing.

| Data | Normal (Non-Bleeding) | Bleeding | Total |
|---|---|---|---|
| Training | 1940 | 1001 | 2941 |
| Testing | 224 | 130 | 354 |
| Total | 2164 | 1131 | 3295 |

The proposed model's architecture is shown in Figure 8. We trained our model to recognize bleeding versus normal (non-bleeding) medical images. The 3295 images in the data set were split into training and validation sets (randomly assigned). The 3295 non-bleeding and bleeding images were separated into 2941 images for training and 354 for validation. Of the images used for training, 1001 were bleeding and 1940 were non-bleeding. Of the images used for validation, 130 images were bleeding and 224 were non-bleeding. Sample images from the validation part of the data set are depicted in Figure 9. The model provided the output for the medical images, assigning them as bleeding or normal (non-bleeding). Table 3 shows the prediction probabilities for the non-bleeding and bleeding classes that were calculated for a few of the sample non-bleeding and bleeding images. The evaluation of the model was performed by comparing the predictions generated by the model with the manually annotated masks, which were approved by a trained physician.

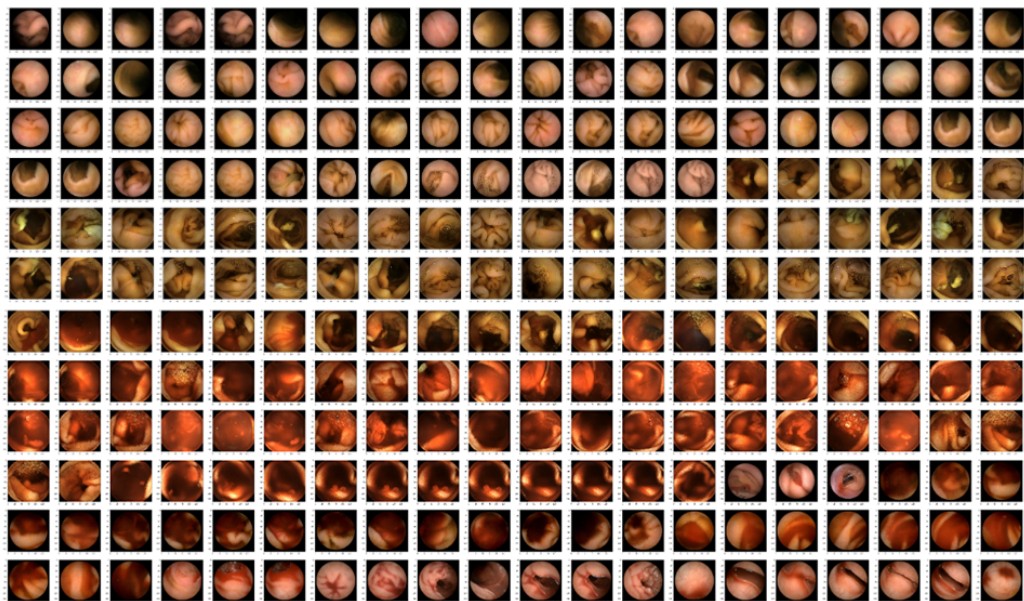

**Figure 9.** Used image data set. Validation part of the bleeding and non-bleeding images.

We proceeded with the images classified as non-bleeding or bleeding from the validation data set. In addition, we implemented three different interpretable Explainable AI algorithms and thereby provided three different explanations for the images. We then conducted a user study to see how the explanations generated by LIME and SHAP affect human decision making.

*5.3. Explainability*

We decided to use three different explainable methods: Local Interpretable Model Agnostic Explanations (LIME), Shapley Additive exPlanations (SHAP) and Contextual Importance and Utility (CIU). We implemented the LIME and SHAP explainable methods on the Triton high-performance computing cluster provided by Aalto University using Python language, whereas CIU explanations were generated using RStudio version 1.2.1335. Our Python implementation code of LIME and SHAP explainability methods is currently open source and publicly available (https://github.com/salujarohit/XAI-for-red-lesions-detection-in-Endoscopy-images, accessed on 4 June 2019), and the same applies to the R implementation of the CIU method for explaining image classifications [42] (https://github.com/KaryFramling/ciu.image, accessed on 4 June 2019). In addition to using explainable methods, we also included a setting without explanations. In this setting, no explanation of any type was given for the displayed images. For our empirical evaluation, we used the black-box XAI as a baseline.

**Table 3.** Prediction probabilities for non-bleeding and bleeding classes for a few of the sample validation images.

| Bleeding Images | Prediction Probability |
|---|---|
| Image 1 | $[0.00 \times 10^0, 1.00 \times 10^0]$ |
| Image 2 | $[0.00 \times 10^0, 1.00 \times 10^0]$ |
| Image 3 | $[0.00 \times 10^0, 1.00 \times 10^0]$ |
| Image 4 | $[0.00 \times 10^0, 1.00 \times 10^0]$ |
| **Non-Bleeding Images** | **Prediction Probability** |
| Image 5 | $[1.00 \times 10^0, 4.61 \times 10^{-31}]$ |
| Image 6 | $[1.00 \times 10^0, 3.78 \times 10^{-24}]$ |
| Image 7 | $[1.00 \times 10^0, 2.82 \times 10^{-32}]$ |
| Image 8 | $[1.00 \times 10^0, 4.28 \times 10^{-29}]$ |

The figures below visualize the explanations of the explainable methods for particular images. Explanations outline the key features (bleeding or non-bleeding areas) in the images, generated by using the interpretable algorithms LIME, SHAP and CIU. With confirmation from a professional physician and manually annotated masks by trained physicians, we ensured that the displayed explanations provided by the algorithms are indeed at least partly marking the correct areas and can be regarded as understandable.

Explanations are given in the form of yellow outlined boundaries around key features of the images that influenced the black-box model's decision.

5.3.1. LIME Explanations

LIME explanations were generated with the following settings: num_samples (size of the neighborhood to learn the linear model) of 2500 and num_features (maximum number of features present in explanation; number of superpixels to include in explanation) of 10. LIME was tested for all images in the validation data set, both bleeding and non-bleeding. Figure 10 depicts some of the explanations provided by LIME. Explanations are given in the form of yellow outlined boundaries around key features of the images that influenced the black-box model's decision. In the case of the bleeding image, the LIME explanation marks the area that positively contributes to the bleeding class, and in the non-bleeding image, the LIME explanation marks the area contributing to the non-bleeding class.

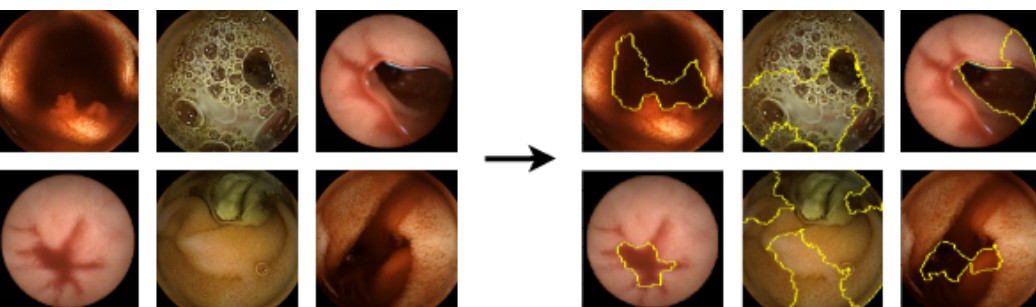

**Figure 10.** LIME explanations.

5.3.2. SHAP Explanations

SHAP was also tested for all images in the validation data set, both bleeding and non-bleeding, and the explanations provided by SHAP are depicted in Figure 11. We applied the model-agnostic Kernel SHAP method on a superpixel segmented image to explain the convolutional neural network's image predictions. SHAP explanations were generated at a num_samples (size of the neighborhood to learn the linear model) value of

2500. In each example, the SHAP explanation of the image depicts contributions to both bleeding and non-bleeding classes. Important features of the image that support the class (bleeding, non-bleeding) are marked in green, and red represents a contradiction to the class (bleeding, non-bleeding).

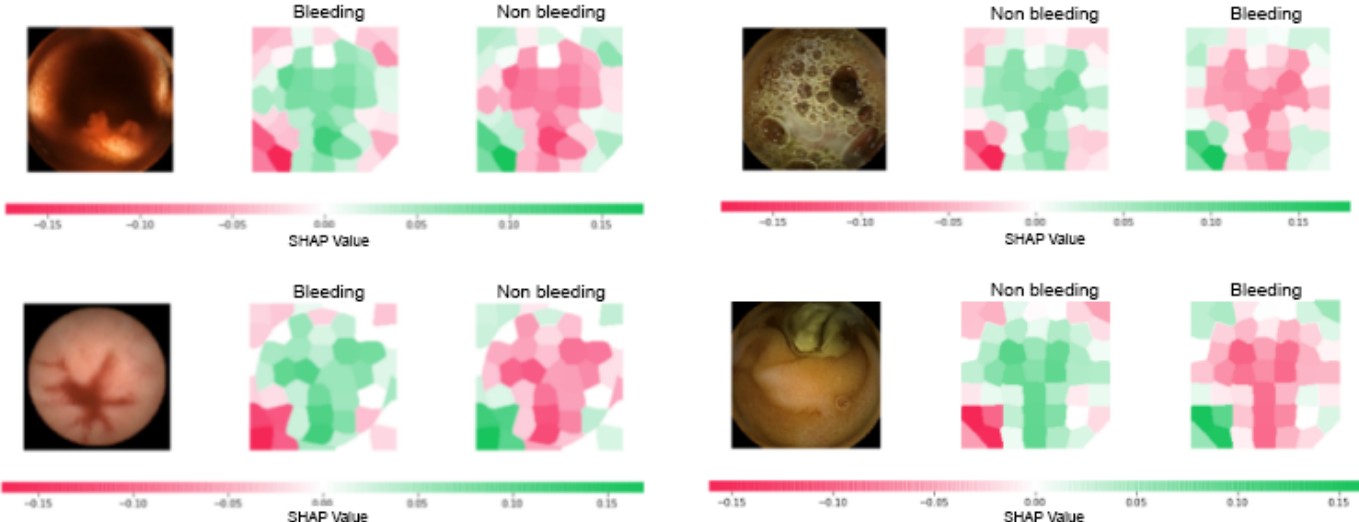

**Figure 11.** A few examples of SHAP explanations produced from the same input data as in the cases of LIME and CIU.

### 5.3.3. CIU Explanations

The CIU explanations were generated at a threshold value of 0.01 and 50 superpixels. CIU was also tested for all images in the validation data set, both bleeding and non-bleeding, and the CIU explanations are shown in Figure 12. CIU explanations are similar to those of LIME, marking the important area in the image that contributes to the given class, either bleeding or non-bleeding.

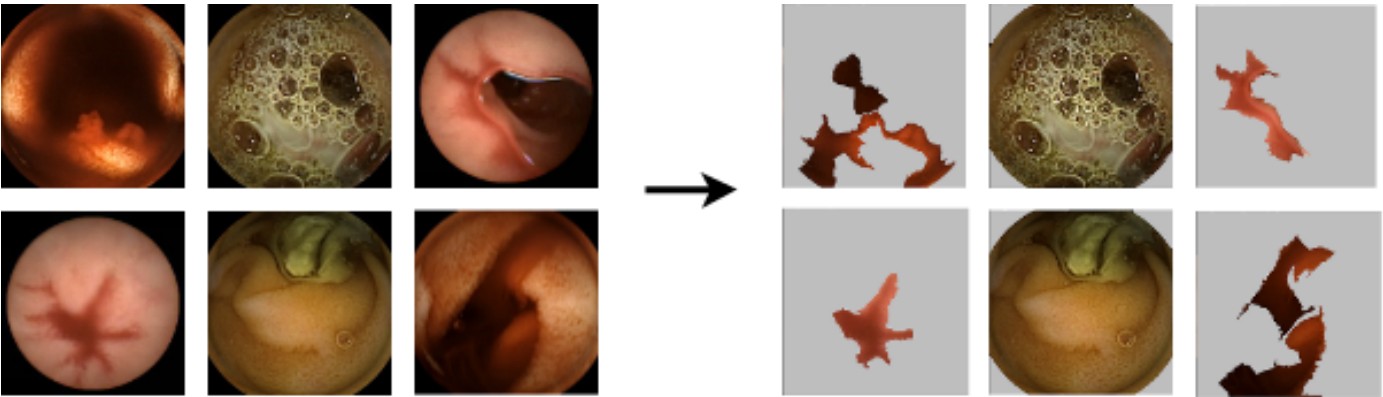

**Figure 12.** CIU explanations.

## 6. Human Evaluation User Study

To investigate the impact of the explainable machine learning methods on human decision-making, we conducted three user studies based on the three explainable methods (LIME, SHAP and CIU) that provided explanations of the images in the medical image data set. Our aim was to see how well users understood the explainable decision-making support and to compare the utility of various explainable methods. We also intended to analyze user satisfaction with the explanations and determine if they could recognize the effectiveness of the explanations.

*6.1. Data Collection*

In order to generalize the use of explainable algorithms, we decided to invite normal users instead of experienced medical professionals. The reason for this is that explanations should be made as simple as possible in order to make them easily understandable even to non-medical people. The users (in this case, normal users) completed a series of tests in a web-based survey and stated their experience and understanding of the presented explanations.

For these user-centered studies, we gathered participants from the university's academic environment, meaning that most of the participants have at least a bachelor's university degree. We collected data from a total of 60 ($n = 60$) users, 20 users in each of the three test groups: 20 users performed testing with no explanation and LIME explanation support, 20 users performed testing with no explanation and SHAP explanation support and 20 users performed testing with no explanation and CIU explanation support. Table 4 shows the demographics of the study participants. The users predominantly have a master's education and STEM (science and technology) background, are in their 20s or 30s and are predominantly males. Approximately half of the participants had heard of XAI prior to participating in the user study.

*6.2. Study Description and Design of the User Study*

Our participants were distributed into three different groups: The first group was presented with noXAI and LIME explanations, the second one with noXAI and SHAP explanations and the third group with noXAI and CIU explanations. The user studies designed for each of the three XAI methods are depicted in Figures 13–15. The basic layout design for the user study is depicted in Figure 16. Figures 17–19 depict the third part of the test phase in which incorrect explanations are also presented.

**Hypotheses.** The aim of this research was to assess the following hypotheses:

**Hypothesis 1 (H1).** *Participants in the first group will perform better when given LIME explanations.*

**Hypothesis 2 (H2).** *Participants in the second group will perform better when given SHAP explanations.*

**Hypothesis 3 (H3).** *Participants in the third group will perform better when given CIU explanations.*

**Hypothesis 4 (H4).** *Participants given CIU explanations will perform better (make more correct decisions) than participants given LIME explanations.*

**Hypothesis 5 (H5).** *Participants given CIU explanations will perform better (make more correct decisions) than participants given SHAP explanations.*

**Hypothesis 6 (H6).** *Participants given CIU explanations will understand the explanations better (distinguish correct explanations from incorrect) than participants given LIME explanations.*

**Hypothesis 7 (H7).** *Participants given CIU explanations will understand the explanations better (distinguish correct explanations from incorrect) than participants given SHAP explanations.*

**Table 4.** Demographics of study participants from LIME, SHAP and CIU user studies (all with included noEXP testing).

| Methods | Total | Gender | | | Highest Degree | | | STEM Background | | XAI Understanding | | Age (years) |
|---|---|---|---|---|---|---|---|---|---|---|---|---|
| | | Female | Male | OTH | Ph.D (or Higher) | Master's Degree | Bachelor's Degree | Yes | No | Yes | No | |
| LIME (and noEXP) | 20 | 6 | 14 | 0 | 3 | 12 | 5 | 19 | 1 | 12 | 8 | 22, 23, 24, 25(2), 26, 27(4), 28, 29, 30(2), 31(3), 32, 33, 34 |
| SHAP (and noEXP) | 20 | 7 | 13 | 0 | 6 | 12 | 2 | 18 | 2 | 8 | 12 | 22(2), 23, 24, 25, 26, 27(4), 28, 29(2), 30, 31, 33, 36, 38, 39, 42, |
| CIU (and noEXP) | 20 | 7 | 13 | 0 | 5 | 9 | 6 | 17 | 3 | 9 | 11 | 21, 25(2), 26(2), 27(3), 28(3), 29(3), 30(2), 32(2), 34, 35 |

**Case 5**: Image 5 with provided explanation:

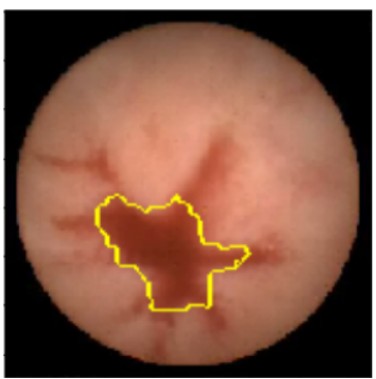

* 21. Is the presented image showing a bleeding or non bleeding section?

○ Bleeding

○ Non bleeding

**Figure 13.** LIME explanations in the user study.

**Case 5**: Image 5 with provided explanation:

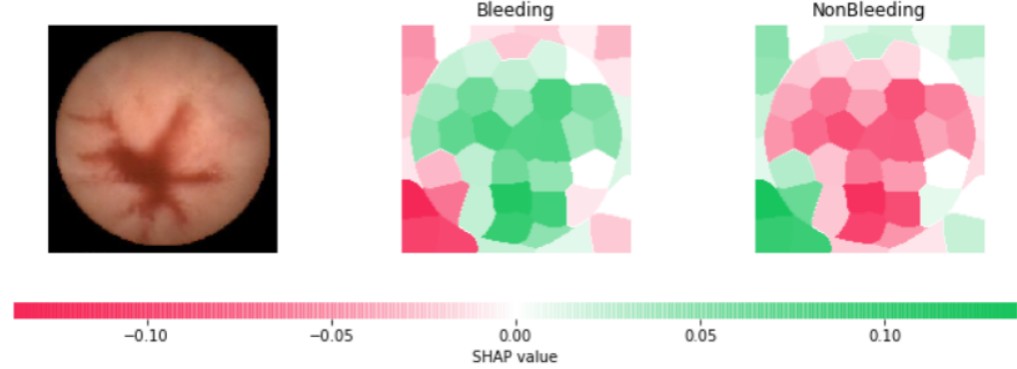

* 21. Is the presented image showing a bleeding or non bleeding section?

○ Bleeding

○ Non bleeding

**Figure 14.** SHAP explanations in the user study.

**Case 5**: Image 5 with provided explanation:

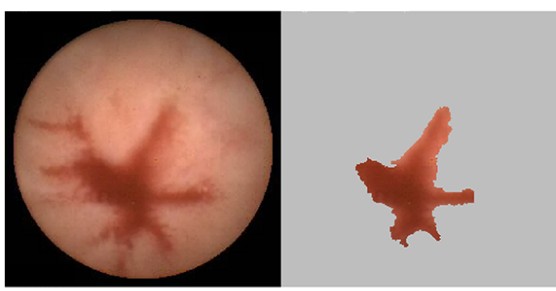

* 21. Is the presented image showing a bleeding or non bleeding section?

○ Bleeding

○ Non bleeding

**Figure 15.** CIU explanations in the user study.

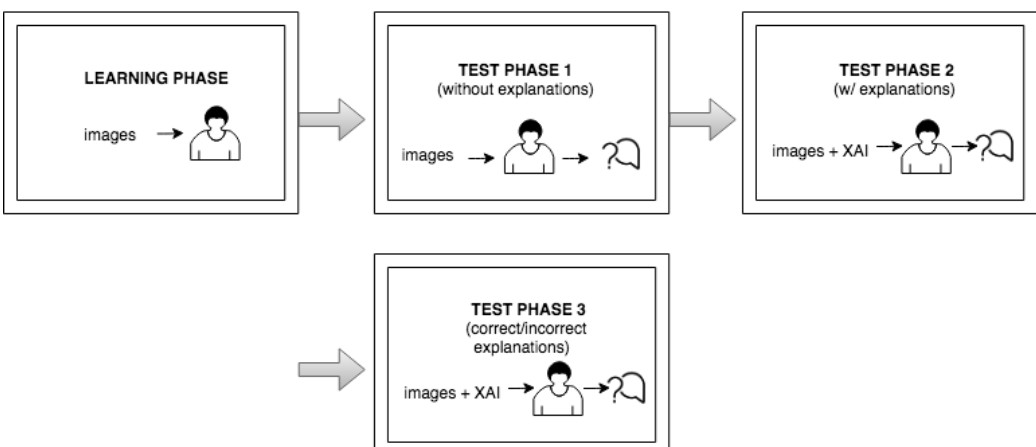

**Figure 16.** User study design.

**Case 1**: Image 1 with provided explanation.
**Prediction**: Image is displaying a **bleeding** section. 💬 0

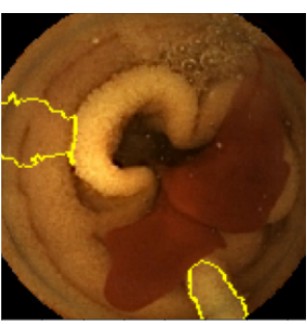

\* 34. Do you think that the explanation accompanying the image is correctly supporting the given prediction?
💬 0

○ Yes

○ No

\* 35. How good do you think the explanation on the image is in supporting the given prediction? 💬 0

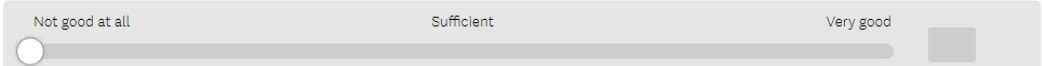

**Figure 17.** Incorrect LIME explanations in the user study.

**Case 1**: Image 1 with provided explanation.
**Prediction**: Image is displaying a **bleeding** section.

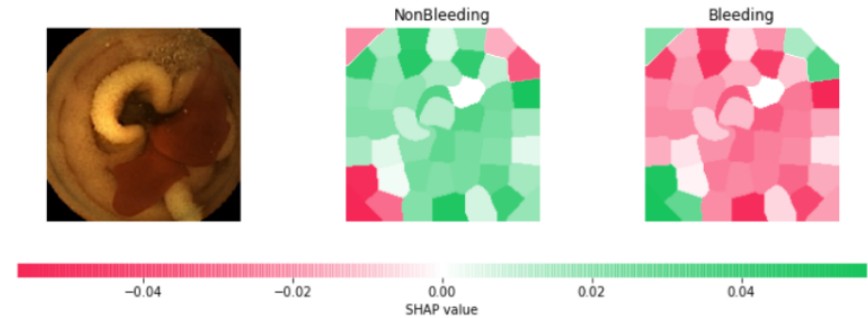

\* 34. Do you think that the explanation accompanying the image is correctly supporting the given prediction?

○ Yes

○ No

\* 35. How good do you think the explanation on the image is in supporting the given prediction?

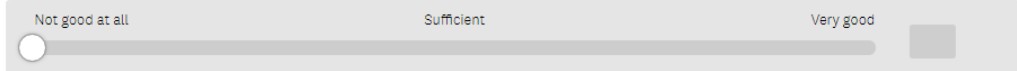

**Figure 18.** Incorrect SHAP explanations in the user study.

**Case 1**: Image 1 with provided explanation.
**Prediction**: Image is displaying a **bleeding** section. ⚲ 0

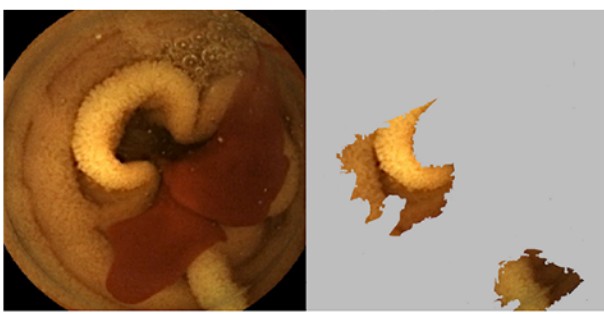

* 34. Do you think that the explanation accompanying the image is correctly supporting the given prediction?
⚲ 0

○ Yes

○ No

* 35. How good do you think the explanation on the image is in supporting the given prediction? ⚲ 0

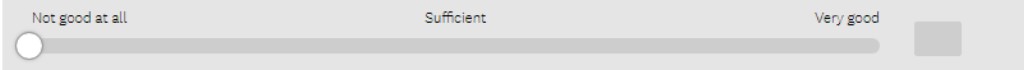

**Figure 19.** Incorrect CIU explanations in the user study.

Tests were performed to determine whether we could reject the null hypotheses (Ha0, Hb0, Hc0, Hd0, He0, Hf0, Hg0), the negations to our seven hypotheses. The first three hypotheses focus on whether explanations support humans in making the correct decisions compared to having no explanation, and the fourth to seventh hypotheses focus on the differences between the three different explanation support methods. The fourth and fifth hypotheses focus on differences in decision-making between users who worked with different explanation support methods. The aim of the last two hypotheses is to evaluate whether human users are able to detect errors in the explanations provided in 5 out of 12 test cases in the last part of the test phase and to assess how the ability to recognize correct or incorrect explanations differs among the three user groups.

1. In the first stage, we presented the selected medical images to the study participants and provided them with the essential information, both verbally and in the form of written instructions, for completing the study.
2. After the users were familiarized with the required instructions, the user study was carried out under the supervision of one of the researchers, who was responsible for the control of the study process.
3. The user study started with the learning part, in which the user was presented with a few test medical images with the model's output so that the user could learn to distinguish bleeding from normal (non-bleeding) images.
4. In the test phase, the user was first presented with medical images that differed from those used in the learning section and had to make a decision about whether the images displayed a bleeding or normal section. In the next phase, the user was presented with the same medical images as in the previous phase, but this time, they were also provided with the visual explanations generated by an explainable

method, without the precise decision offered by a black-box model. The explanation was presented in such a way that the important features in the initial image from our data set (in this case, normal or bleeding areas, if any) were highlighted or isolated. The user was instructed to make the diagnosis in both test phases. The user needed to decide if there was any bleeding present and if the image showed a severe condition or if there was no bleeding present. Thus, we could verify whether the proposed explainable methods enhanced the number of correct decisions made by the human. At the end of the second test phase, the user had to rate the explanations by marking how satisfied they were with them on a Likert scale ranging from 0 to 5 (where 0 indicates the lowest satisfaction with the explanations, and 5 indicates the highest satisfaction).

5. In the last part of the test phase, users had to indicate if they thought that the presented explanation was correct or not for each presented image. Thus, we collected information about users' ability to understand the explanations by having them judge whether the explainable method provided a correct or incorrect explanation.

6. The process was iterated for four different cases of data in the learning stage, 16 cases in the second stage and 12 cases in the third stage. Cases in the learning phase differed from those in the test phase. The presented images from the first and second test phases also differed from those in the third part of the test phase. In the test phase, users were not allowed to reference the learning data.

7. After all four rounds of the survey were completed, the users were also asked to complete the evaluation questionnaire, through which we collected information on their demographics and their understanding of the explanation support. Because these questions could influence the users' perception of the procedure, they were only posed after the research evaluation was completed and could not be accessed by the user beforehand.

Throughout the trial, the same data points were used for all participants in each user group. This architecture accounts for any data discrepancies between conditions and users. We also matched the data across the no-explanation and explanation phases to control for the impact of specific data points on user accuracy. In order to isolate the impact of the explanation support, we measured the users' initial accuracy prior to measuring the accuracy when they were given the help of the explanations.

## 7. Analyses of the Results

### 7.1. Performance of LIME, SHAP and CIU

When generating explanations with the three different explainable methods, we also compared the time needed to generate explanations and their overall performance, as shown in Table 5. LIME needed around 11 s per image (11.4 s) and around 5 min and 20 s for 28 images with num_samples = 2500 and num_features = 10. For generating explanations on all the validation images (354 images), the time needed was 1 h and 45 min. SHAP needed around 10 s per image (9.8 s) and around 4 min and 35 s for 28 images with num_samples = 3000. This would be around 1 h 30 min for all validation images. The running time of the CIU explainable method was less than that of SHAP and LIME; CIU required about 8.5 s per image, and the total time for producing explanations for 28 selected images from our data set was 4 min. For all validation images, the timing would be around 1 h and 18 min.

**Table 5.** LIME, SHAP and CIU time comparison (note: LIME and SHAP were run using Python and CIU was run using R).

|  |  | **LIME** | **SHAP** | **CIU** |
|---|---|---|---|---|
|  | 1 image | 11.40 s | 9.80 s | 8.50 s |
| Time comparison | 28 images | 5 min 20 s | 4 min 30 s | 4 min |
|  | 354 images | 1 h 45 min | 1 h 30 min | 1 h 18 min |

Note that experiments for LIME and SHAP were run using Triton, a high-performance computing cluster provided by Aalto University, whereas experiments for CIU were run using RStudio version 1.2.1335 (R version 3.6.1) on a MacBook Pro, with a 2.3 GHz 4-Core Intel Core i7 processor, 16 GB 1333 MHz DDR3 memory and Intel HD Graphics 3000 512 MB graphics card. When running LIME in RStudio, it took about 1 min and 50 s to generate explanations for each image. SHAP was not tested using RStudio.

*7.2. Quantitative Analyses of Explanations*

In order to explore the impact of explanations generated by the three explainable methods on human decision-making, we first analyzed results for each group and compared the difference between users' performance with and without explanations generated by one of the explainable methods. After that, we compared the three different user study groups: those who were given (1) LIME explanations, (2) SHAP explanations and (3) CIU explanations. We used IBM SPSS Statistics version 23.0.0.0 to analyze the data, run hypothesis tests and compute exploratory statistics. When analyzing the data from the three user study settings, we first examined the differences between means and medians of human decision making (Tables 6–9); for each of the hypotheses tested, the difference in human decision making was evaluated using independent-sample *t*-test assuming unequal variances with a significance level of $\alpha$ set to 0.05 (Table 10).

**Table 6.** Mean and median values of LIME users' decision-making.

|  | **Measures** | **LIME User Study** | |
|---|---|---|---|
|  |  | **With Explanation** | **Without Explanation** |
| Correct decision | Mean | 14.15 | 13.95 |
|  | Median | 14.50 | 15.00 |
| Incorrect decision | Mean | 1.80 | 2.05 |
|  | Median | 1.50 | 1.00 |

**Table 7.** Mean and median values of SHAP users' decision-making.

|  | **Measures** | **SHAP User Study** | |
|---|---|---|---|
|  |  | **With Explanation** | **Without Explanation** |
| Correct decision | Mean | 13.40 | 14.05 |
|  | Median | 15.00 | 14.00 |
| Incorrect decision | Mean | 2.60 | 1.95 |
|  | Median | 1.00 | 2.00 |

**Table 8.** Mean and median values of CIU users' decision-making.

| | Measures | CIU User Study | |
|---|---|---|---|
| | | **With Explanation** | **Without Explanation** |
| Correct decision | Mean | 14.90 | 14.30 |
| | Median | 16.00 | 14.00 |
| Incorrect decision | Mean | 1.10 | 1.70 |
| | Median | 0.00 | 2.00 |

**Table 9.** Mean and median values of users' ability to recognize correct and incorrect explanations.

| | Measure | LIME | SHAP | CIU |
|---|---|---|---|---|
| Recognition of correct and incorrect explanations | Mean | 8.85 | 8.65 | 10.25 |
| | Median | 9.50 | 9.50 | 11.00 |

**Table 10.** Hypothesis analyses (note: * $p < 0.05$, ** $p < 0.01$) .

| | *t*-Test | Hypothesis | *p*-Value (One-Tailed) | *p*-Value (Two-Tailed) |
|---|---|---|---|---|
| 1 | (LIME, noEXP) | Ha0 | 0.334 | 0.738 |
| 2 | (SHAP, noEXP) | Hb0 | 0.232 | 0.464 |
| 3 | (CIU, noEXP) | Hc0 | 0.079 | 0.158 |
| 4 | (CIU, LIME) | Hd0 | 0.059 | 0.120 |
| 5 | (CIU, SHAP) | He0 | 0.036 * | 0.073 |
| 6 | (CIU, LIME) | Hf0 | 0.009 ** | 0.018 * |
| 7 | (CIU, SHAP) | Hg0 | 0.037 * | 0.073 |

7.2.1. Analyses of Human Decision-Making in the Three Users Groups with Different Explanation Support Methods

Tables 6–8 show the mean and median of correct and incorrect decisions for each type of explanation as well as the no-explanation setting for all three user studies. Aligned with our first five hypotheses, there are notable differences in the means regarding differences in the decision-making of users between different explanation settings, as well as between settings with and without explanations. Table 9 shows notable differences in means in relation to our sixth and seventh hypotheses regarding differences in users' understanding of explanations based on whether they distinguished incorrect explanations from correct ones. The analysis of all seven hypotheses is shown in Table 10. Additionally, Table 11 shows the difference between users given LIME explanations and those given SHAP explanations.

**Table 11.** Comparison between LIME and SHAP.

| | *t*-Test | | *p*-Value (One-Tailed) | *p*-Value (Two-Tailed) |
|---|---|---|---|---|
| 1 | (LIME, SHAP) | User's decision making | 0.185 | 0.370 |
| 2 | (LIME, SHAP) | Recognition of correct and incorrect explanations | 0.414 | 0.827 |

**LIME.** The results for users provided LIME explanation support show that, compared to the setting without explanations, participants performed slightly better in the test phase with the explanations provided, which is in line with our first hypothesis. Although the

difference was not statistically significant (*p* = 0.738) relative to testing without explanations, users answered more questions correctly (Tables 6 and 10) in the test phase with LIME explanation support, meaning that they recognized that the image displayed a bleeding or non-bleeding sequence at a higher frequency. The results from the third test phase show that, on average, participants were also able to recognize when the displayed LIME explanation was correct or incorrect. The mean of the correct decision was 8.85 out of 12 answers in total (Table 9).

**SHAP.** The results for SHAP show that users answered more questions correctly in tests without explanations, meaning that they recognized that the image displayed a bleeding or non-bleeding section at a higher frequency, although the difference was not statistically significant (*p* = 0.464) relative to testing with explanations (Tables 7 and 10). On the other hand, on average, participants were able to recognize when the displayed SHAP explanation was correct or incorrect. The mean of the correct answers was 8.47 out of 12 answers in total (Table 9).

**CIU.** In line with our third hypothesis, the results for CIU show that users answered more questions correctly in the test phase with explanation support, meaning that they recognized that the image displayed a bleeding or non-bleeding condition at a higher frequency, although the difference was not statistically significant (*p* = 0.158) relative to testing without explanations (Tables 8 and 10). On average, participants were very good at recognizing when the displayed CIU explanation was correct or incorrect. The mean of the correct answers was 10.25 out of 12 answers in total (Table 9).

**Comparison between LIME and SHAP.** We performed a between-group comparison in order to determine which explainable method (LIME, SHAP or CIU) was associated with better decision making by users. When comparing the performance between LIME and SHAP, users who received SHAP explanations reported higher satisfaction with the explanations, with the difference being statistically significant (*p* = 0.0135, Tables 12 and 13). Similarly, users who received SHAP explanations reported a greater understanding of explanations compared to users provided with LIME explanations, although the difference was not statistically significant (*p* = 0.999, Tables 12 and 13). However, users given LIME explanation support answered more questions correctly than users who received SHAP support, although the difference was not statistically significant (*p* = 0.370, Tables 6, 7 and 11). Users given SHAP explanations also required significantly more time to complete the study compared to those provided with LIME explanation support (*p* = 0.011, Tables 12 and 13), which may indicate that SHAP explanations required more in-depth concentration and were harder to interpret. When comparing users who were able to distinguish incorrect explanations from correct ones, the users given LIME explanation support performed better when compared to the users who received SHAP explanations, although the difference was not statistically significant (*p* = 0.827, Table 11).

**Table 12.** Mean and median values of satisfaction, understanding and time spent.

|  |  | LIME | SHAP | CIU |
|---|---|---|---|---|
| Satisfaction | Mean | 2.00 | 3.20 | 3.75 |
|  | Meadian | 2.00 | 3.00 | 4.00 |
| Time in minutes | Mean | 15.57 | 23.18 | 16.30 |
|  | Median | 14.83 | 21.18 | 15.73 |
| Understanding | Yes | 16.00 | 16.00 | 18.00 |
|  | No | 4.00 | 4.00 | 2.00 |

**Table 13.** *t*-tests: Satisfaction, understanding and time spent (note: * $p < 0.05$, ** $p < 0.01$, *** $p < 0.001$).

|  | *t*-Test | *p*-Value (One-Tailed) | *p*-Value (Two-Tailed) |
|---|---|---|---|
| Satisfaction | (LIME, SHAP) | 0.007 ** | 0.0135 * |
|  | (LIME, CIU) | 0.000 *** | 0.000 *** |
|  | (SHAP, CIU) | 0.072 | 0.144 |
| Understanding | (LIME, SHAP) | 0.499 | 0.999 |
|  | (LIME, CIU) | 0.195 | 0.389 |
|  | (SHAP, CIU) | 0.195 | 0.389 |
| Time in minutes | (LIME, SHAP) | 0.005 ** | 0.011 ** |
|  | (LIME, CIU) | 0.317 | 0.633 |
|  | (SHAP, CIU) | 0.008 ** | 0.0166 ** |

**Comparison between CIU, LIME and SHAP.** In comparison with users given LIME or SHAP explanation support, users given CIU explanation support answered more questions correctly, although the difference was not significant ($p = 0.120$; $p = 0.073$, Tables 6–8 and 10). In terms of the time required to complete the study, users given CIU explanation support spent significantly less time than those given SHAP explanations ($p = 0.016$, Tables 12 and 13) and more time than users given LIME explanations ($p = 0.633$, Tables 12 and 13), but in this case, the time difference was not statistically significant. When comparing the ability of users to distinguish between correct and incorrect explanations, the users given CIU explanation support showed a better understanding of the explanations than both users given LIME ($p = 0.120$, Tables 9 and 10) and SHAP explanation support ($p = 0.073$, Tables 9 and 10), although the difference was not statistically significant. In addition, participants given CIU explanations answered more questions correctly when provided with explanations compared to the no-explanation condition, although, again, the difference was not statistically significant ($p = 0.158$, Tables 6–8 and 10). Users also reported significantly higher satisfaction with CIU explanations compared to users who received LIME explanations ($p = 0.000$, Tables 12 and 13) and higher but not statistically significant satisfaction compared to users provided with SHAP explanations ($p = 0.144$, Tables 12 and 13). Similarly, users who received CIU explanations reported having a greater understanding of the explanations in comparison to those provided with LIME explanations ($p = 0.389$, Table 13), as well as a better understanding than users who received SHAP explanations ($p = 0.389$, Table 13), although the difference was not statistically significant.

### 7.2.2. Correlation Analyses

We also performed correlation analyses between users' performance (count of correct decisions) and different demographic variables (age, gender, education level, STEM background and knowledge of XAI), as well as between their performance and the time spent completing the study and the users' understanding and satisfaction with the explanations. The correlations were calculated using Spearman's rank correlation coefficient. Spearman's correlation was chosen because it captures the monotonic relationship between the variables instead of only a linear relationship, and it also works well with categorical variables such as gender. Spearman's correlation coefficient values for all conditions of LIME, SHAP and CIU explanation support, together with p-values, are shown in Table 14. Table 15 shows correlation analyses for the settings without explanations.

**LIME.** In the case of LIME, we found a significant correlation between the number of correct decisions by users and their age ($p = 0.046$, Table 14), with younger participants making a higher number of correct decisions, and satisfaction with the provided explanations ($p = 0.019$, Table 14). Interestingly, the lower the users' satisfaction, the better the

users were in making decisions, which may indicate that the users who recognized LIME support in decision-making also recognized that it could do a better job in decision support. In the setting without explanations, we did not find any significant correlations (Table 15).

**SHAP.** In the setting with SHAP explanation support, we found correlations between users' number of correct decisions and higher satisfaction with SHAP explanations ($p = 0.031$, Table 14) and statements that they understood SHAP better ($p = 0.018$, Table 14). This may indicate that users in general were satisfied with SHAP explanations and their presentation; however, because the presentation of SHAP explanations was more complex, users appeared to have a harder time understanding them. In the setting without explanations, we found significant correlations between the number of correct decisions by users and less time needed to complete the study (Table 15).

**CIU.** We did not find any significant correlations in the setting with CIU explanation support or in the setting without the provision of explanations (Tables 14 and 15).

**Table 14.** Correlation between demographic variables and decision-making in XAI settings (note: * $p < 0.05$).

| Variable | | LIME | SHAP | CIU |
|---|---|---|---|---|
| Age | correlation | −0.451 | 0.229 | −0.413 |
| | *p*-Value (two-tailed) | 0.046 * | 0.332 | 0.071 |
| Gender | correlation | 0.349 | 0.272 | −0.02 |
| | *p*-Value (two-tailed) | 0.132 | 0.246 | 0.934 |
| Education | correlation | −0.073 | 0.210 | −0.321 |
| | *p*-Value (two-tailed) | 0.760 | 0.374 | 0.167 |
| STEM background | correlation | 0.387 | −0.309 | −0.132 |
| | *p*-Value (two-tailed) | 0.092 | 0.185 | 0.580 |
| XAI | correlation | −0.235 | 0.274 | −0.435 |
| | *p*-Value (two-tailed) | 0.318 | 0.242 | 0.055 |
| Time spent | correlation | −0.271 | 0.480 | −0.10 |
| | *p*-Value (two-tailed) | 0.248 | 0.840 | 0.674 |
| Satisfaction | correlation | −0.519 | 0.482 | 0.396 |
| | *p*-Value (two-tailed) | 0.019 * | 0.031 * | 0.084 |
| Understanding | correlation | −0.377 | 0.522 | 0.188 |
| | *p*-Value (two-tailed) | 0.101 | 0.018 | 0.427 |

**Table 15.** Correlation between demographic variables and decision making in noEXP settings (note: * $p < 0.05$, two-tailed).

| Variable | | noEXP (LIME) | noEXP (SHAP) | noEXP (CIU) |
|---|---|---|---|---|
| Age | correlation | −0.18 | −0.063 | −0.318 |
| | *p*-Value | 0.447 | 0.792 | 0.171 |
| Gender | correlation | 0.069 | −0.166 | 0.313 |
| | *p*-Value | 0.773 | 0.484 | 0.179 |
| Education | correlation | 0.173 | −0.246 | −0.087 |
| | *p*-Value | 0.465 | 0.296 | 0.715 |
| STEM | correlation | 0.392 | 0.217 | 0.139 |
| | *p*-Value | 0.087 | 0.357 | 0.558 |
| XAI | correlation | 0 | −0.285 | −0.264 |
| | *p*-Value | 1 | 0.223 | 0.261 |
| Time spent | correlation | −0.191 | −0.491 | −0.191 |
| | *p*-Value | 0.419 | 0.028 * | 0.419 |

### 7.3. Qualitative Analyses of Explanations

Most of the participants provided with LIME, SHAP or CIU explanations reported that they were able to understand the provided CIU explanations, which, as shown in the Tables 12 and 13, received the highest rating. User ratings of the explanations show that they were most satisfied with the explanations generated by CIU and less so with those of SHAP or LIME. They were mostly satisfied with the provided explanations but also noted that they could be more precise and should cover a larger area of the image. By analyzing the participants' feedback statements from the evaluation questionnaire, we also found that:

1. Users want more precise identification of the important areas in some of the presented images.
2. In addition to visual explanations, users want supplementary text explanations.
3. Users would like to have an option to interact with the explainable method in order to gain more in-depth information.

## 8. Discussion

In the present study, we observed notable differences in human decision-making between three groups of users who worked with different explanation support methods. The observed differences reflect our initial assumption that users who receive CIU explanation support would perform better than those provided with SHAP or LIME explanation support. Our results suggest that CIU was more helpful to users in making the correct decisions, who were also more satisfied with its presentation compared to users receiving LIME or SHAP support. This indicates that explanations generated by CIU were clearer to the users and thus provided greater support in decision-making.

Our results also support the last two hypotheses, which state that participants with CIU will better understand the provided explanations and thereby better distinguish between correct and incorrect explanations compared to participants receiving LIME or SHAP explanation support. Users with CIU explanation support were significantly better at recognizing incorrect explanations than those given LIME explanations and, to some extent, better than those provided with SHAP explanation support. Additionally, users given CIU explanation support spent significantly less time completing the user study than users given SHAP explanations and more time than those given LIME explanations, but in this case, the time difference was not statistically significant. A possible explanation for

this result is that participants were able to better understand the explanations provided by CIU because they are depicted with lower complexity compared to those of SHAP, and the representation of the significant area in the images is more accurate with CIU than with LIME. This assumption is also supported by users' statements regarding their satisfaction and understanding of the explanation support.

The present results also provide insight into the initial research questions concerning the use of XAI methods as human decision support and their contribution to the increased trust in AI-based computer vision systems in the medical domain. The user studies and testing performed with explanations compared to those without them suggest that users are better at decision-making when assisted by explainable methods and are comfortable with the explanations provided. In two out of three user studies (LIME and CIU explanation support), the participants performed better when the explanation support was provided. However, this was not the case for the users provided with SHAP explanations. Although users given SHAP explanations were relatively good at recognizing correct and incorrect explanations, the explanations by SHAP did not prove to increase the number of correct decisions compared to the no-explanation setting. In comparison to the user studies with LIME and CIU, the number of correct answers in both settings with SHAP was relatively low; however, the setting without explanations proved to increase the number of correct decisions by users, which may indicate that SHAP explanations were harder to understand and were confusing to the users.

### 8.1. Limitations

The present paper provides an evaluation of explainable methods (LIME, SHAP and CIU) for supporting human decision-making. However, this study has a set of limitations, with the most important ones listed below.

1. The current study's focus is limited to a single medical data set. The current use of explanation support focuses only on one set of medical images, which can be further tested on other more complex medical cases in need of decision support for various diagnoses.
2. It is important to expand the current scope of the studied data and apply these explanation methods to real-life settings. Using the data in real-world scenarios may facilitate their practical application.
3. The scope of evaluation was limited to basic tests with laypersons due to time constraints. The study can be generalized to carry out application-based evaluations involving real tasks performed by domain experts. In the case of medical data, the most suitable users would be physicians working in diagnostics.

### 8.2. Future Work

To overcome the limitations addressed in the previous subsection, the following research directions may be considered in the future.

1. With future improvements, we aim to generalize the explanations provided by explainable methods (LIME, SHAP and CIU) by using different medical data sets and thereby provide greater decision support for medical experts.
2. In this study, the number of participants was limited to 60 (20 for each case). The results should ideally be validated with a larger sample size. In addition, increasing the sample size could help produce more statistically meaningful hypothesis test results.
3. In order to improve the evaluation and test the usability of the explanations, a user evaluation study with domain (medical) experts is required. Furthermore, the explanations could potentially be tested using application-based assessment, which would require domain experts conducting activities related to the use of the explanations.
4. In the future, we aim to expand on the current work by dealing with real-life case scenarios. It would be interesting to work with real-world complexities in order to show that the explainable methods can help people to make better decisions in real-world situations.

## 9. Conclusions

In line with our results, we present three potential explainable methods that, with future improvements in implementation, can be generalized to different medical data sets and can provide effective decision support for medical experts. From the viewpoint of users, our research offers deep insight into the details of explanation support and can be used as constructive feedback for the potential implementation of explainable machine learning methods in the future. Our findings suggest that there are notable differences in human decision-making between various explanation support settings, with CIU showing the best decision support and performance. Additionally, in comparison to the no-explanation setting, explanation support proved to increase the number of correct decisions made by users in two out of three user studies. The presented work can thus give developers more confidence to further develop and utilize explainable methods, which, in turn, will instill users with more confidence and trust. As the explanations were evaluated using laypeople in the present study, in the future, we may evaluate these methods using application-grounded evaluation, which would involve domain experts performing tasks specific to the use of the explanations. Such research could provide a clearer evaluation of the explanations, and with future improvements and utilization, the explanations could be applied to other medical image processing situations. The application of explainable methods to other medical data sets, as well as further testing and improvements in the context of providing decision support to medical professionals and automating diagnostic procedures, may lead to solutions that are more broadly applicable.

**Author Contributions:** Conceptualization, S.K.; methodology, S.K., R.S. and K.F.; software, R.S. and K.F.; validation, A.M. and K.F.; formal analysis, S.K.; investigation, S.K., A.M. and R.S.; resources, A.M.; writing—original draft preparation, S.K.; writing—review and editing, S.K. and A.M.; visualization, S.K.; supervision, A.M. and K.F.; project administration, A.M. and K.F.; funding acquisition, A.M. All authors have read and agreed to the published version of the manuscript.

**Funding:** The research reported in this publication is supported by Helsinki Institute for Information Technology (grant 9160045), under the Finnish Center for Artificial Intelligence (FCAI) unit. The work is partially supported by the Wallenberg AI, Autonomous Systems and Software Program (WASP) funded by the Knut and Alice Wallenberg Foundation and project 856602 "FINEST TWINS" funded by the EU Horizon 2020 program.

**Institutional Review Board Statement:** Not aplicable. The study was conducted according to the guidelines of the Declaration of Helsinki.

**Informed Consent Statement:** Informed consent was obtained from all subjects involved in the study.

**Data Availability Statement:** The used 3295 images in the Red Lesion Endoscopy data set are publicly available and can be retrieved from Coelho (https://rdm.inesctec.pt/dataset/nis-2018-003, accessed on 4 June 2019) [2].

**Conflicts of Interest:** The authors declare no conflict of interest.

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
