# Peer review of "Explainable Artificial Intelligence for Human Decision Support System in the Medical Domain"

_make, doi:10.3390/make3030037_

Round 1

Reviewer 1 Report

Three user groups (n=20, 20, 20) with three distinct forms of explanations were quantitatively analyzed. We have found that, as hypothesized, the CIU explainable method performed better than both LIME and SHAP methods in terms of increasing support for human decision-making as well as being more transparent and thus understandable to users. Additionally, CIU outperformed LIME and SHAP by generating explanations more rapidly. Our findings suggest that there are notable differences in human decision-making between various explanation support settings. In line with that, the authors present three potential explainable methods that can with future improvements in implementation be generalized on different medical data sets and can provide great decision-support for medical experts.

  • This is familiar to our readers. Pls shorten “The well-trained machine learning systems have the potential to make accurate predictions from the individual subjects about various anomalies, and can therefore be used as an effective clinical practice tool. However, even though we can understand their core mathematical concepts, they are considered black-box models which lack an explicit declarative information representation and have trouble producing the underlying explanatory structures”
  • Some related examples could be listed, see “PSSPNN: PatchShuffle stochastic pooling neural network for an explainable diagnosis of COVID-19 with multiple-way data augmentation” and “ANC: Attention network for COVID-19 explainable diagnosis based on convolutional block attention module”
  • More explanation techniques are not discussed at Section 4, such as CAM.
  • The comparison tables are missing.
  • In all, it is a good paper

Author Response

Thank you for your review on our paper “Explainable Artificial Intelligence for Human Decision-Support System in Medical Domain”. Attached we are sending a point by point comment to your suggestions.

Best regards,

Reviewer 2 Report

In this manuscript, the authors have presented the application of explainable AI for the human decision-support system in the medical domain. Overall paper is well written and detailed but it requires some major improvements before it can be accepted for publication.

1) Please write the full form of the abbreviations before using them in the text such as Local Interpretable Model-Agnostic Explanations (LIME) and SHapley Additive exPlanations (SHAP). 

2) Introduction, related work, and background are too much wordy. Please try to make it digestible by including 1-2 more illustrations or tables to explain through.

3) Figure 1 for the basic concept of XAI is included but its explanation is missing. 

4) Quality of some figures is very low and it is difficult to read the text. For e.g. In figure 2, text in the verticle boxes is very hard to read. 

5) Include a comparison table that should give the strengths and weaknesses of different XAI methods that you studied in this article. 

Author Response

(The authors gave the same response as above.)

Reviewer 3 Report

The contribution presents the potential of explainable artificial intelligence methods for decision-support in medical image analysis (here in-vivo gastral images obtained from a video capsule endoscopy). The study has the aim to increase the trust of health professionals in black box predictions. I agree with the authors that the black box behavior, especially of Deep Learning-based approaches, is still a major issue (despite the success of Deep Learning also in the medical domain, providing currently state of the art results) across all medical applications, please see (and cite) the second last paragraph of this preprint meta review in medical deep learning:

https://arxiv.org/abs/2010.14881

Minor issues:

Ribeiro et al -> Ribeiro et al. (dot at et al.)

The quality of the figures is very bad, e.g. the labels of Figure 3 are almost not readable.

The references need to be reworked there are several typos and issues, e.g. [10], [11] and [12].

[2] … ac- cessed …

Author Response

Thank you for your review on our paper “Explainable Artificial Intelligence for Human Decision-Support System in Medical Domain”. Attached we are sending point by point comments to your suggestions.

Best regards,

Round 2

Reviewer 1 Report

Accept in present form.

Reviewer 2 Report

All of my comments are addressed. I recommend the acceptance of the manuscript in its current form.